# Algal Hydrogen Production and Exopolysaccharide Patterns in *Chlorella–Bacillus* Inter-Kingdom Co-Cultures

Bettina Hupp [1,2], Gabriella Huszár [1], Attila Farkas [1] and Gergely Maróti [1,3,*]

1   Institute of Plant Biology, Biological Research Center, Eötvös Loránd Research Network (ELKH),
    Temesvári Krt. 62., H-6726 Szeged, Hungary; huppbettina@gmail.com (B.H.); huszargabb@gmail.com (G.H.);
    farkas.attila@brc.hu (A.F.)
2   Doctoral School in Biology, Faculty of Science and Informatics, University of Szeged, H-6726 Szeged, Hungary
3   Department of Water Sciences, University of Public Service, H-6500 Baja, Hungary
*   Correspondence: maroti.gergely@brc.hu

**Abstract:** Biohydrogen production from wastewater using eukaryotic green algae can be facilitated by appropriately selected bacterial partners and cultivation conditions. Two *Chlorella* algal species were chosen for these experiments, based on their robust growth ability in synthetic wastewater. The applied three *Bacillus* bacterial partners showed active respiration and efficient biomass production in the same synthetic wastewater. *Bacillus amyloliquefaciens*, *Bacillus mycoides*, and *Bacillus cereus* as bacterial partners were shown to specifically promote algal biomass yield. Various inter-kingdom co-culture combinations were investigated for algal–bacterial biomass generation, for co-culture-specific exopolysaccharide patterns, and, primarily, for algal biohydrogen evolution. *Chlorella* sp. MACC-38 mono- and co-cultures generated significantly higher biomass compared with that of *Chlorella* sp. MACC-360 mono- and co-cultures, while in terms of hydrogen production, *Chlorella* sp. MACC-360 co-cultures clearly surpassed their *Chlorella* sp. MACC-38 counterparts. Imaging studies revealed tight physical interactions between the algal and bacterial partners and revealed the formation of co-culture-specific exopolysaccharides. Efficient bacterial respiration was in clear correlation with algal hydrogen production. Stable and sustainable algal hydrogen production was observed in synthetic wastewater for *Chlorella* sp. MACC-360 green algae in co-cultures with either *Bacillus amyloliquefaciens* or *Bacillus cereus*. The highest algal hydrogen yields (30 mL $H_2$ $L^{-1}$ $d^{-1}$) were obtained when *Chlorella* sp. MACC-360 was co-cultured with *Bacillus amyloliquefaciens*. Further co-culture-specific algal biomolecules such as co-cultivation-specific exopolysaccharides increase the valorization potential of algal–bacterial co-cultures and might contribute to the feasibility of algal biohydrogen production technologies.

**Keywords:** algal–bacterial interaction; *Chlorella*; *Bacillus*; biohydrogen; green algae; exopolysaccharide





## 1. Introduction

Increasing environmental pollution and global warming have led to an increased focus on renewable, sustainable and environmentally friendly alternative energy sources, including algae biofuels [1]. Microalgae (including prokaryotic cyanobacteria and eukaryotic unicellular algae) are simple, chlorophyll-containing photosynthetic organisms with diverse biotechnological exploitation potential. Hydrogen is considered a promising clean alternative renewable energy carrier. Microalgae are capable of biohydrogen production through both photolytic and fermentation pathways; the algae cells use their hydrogenase enzymes for the disposal of excess electrons in the form of molecular hydrogen [2–5]. Eukaryotic green algae utilize Fe-Fe hydrogenases for converting the energy of sunlight or organic macromolecules into hydrogen gas [6–9]. The direct and indirect photolytic pathways take advantage of the photosynthetic system of the algae, linking water splitting or starch degradation to hydrogen production [10–12].

Sunlight has primary importance in algae physiology. Microalgae are capable of utilizing light for photosynthesis within the wavelength range of 400 nm to 700 nm [13]. The protein, carbohydrate, lipid, and pigment compositions of green algae are all highly dependent on light conditions. In their natural habitats, algae predominantly grow in diminished light and avoid direct sunlight, which can be harmful to algal cultures [14]. *Chlamydomonas reinhardtii* is the most studied and is a benchmark unicellular green algal strain capable of hydrogen production [10,15–18]. However, the *Chlorella* genus is another well-known robust, high-biomass-producing green algae taxon [19–22]. The *Chlorella* species have the capacity to rapidly accumulate high biomass and to produce phytohormones such as auxins, ethylene, brassinosteroids, cytokinin, and trehalose [23,24], as well as exopolysaccharides [25–27]. Thus, *Chlorella* strains are among the most popular eukaryotic green microalgae due to their versatility and exploitability in various biotechnological industries, including wastewater treatment and food and feed additive production, as well as the pharmaceutical industry [21].

Recently, the use of synthetic algal and microbial co-cultures and engineered consortia have attracted particular interest in the biohydrogen research [5,10,12,28,29]. A number of studies have investigated and compared the efficiency of algal biohydrogen production using either axenic algae or algal–bacterial co-cultures. Generally, microalgae produce oxygen during photosynthetic growth, which can be utilized by bacteria as an electron acceptor in the degradation of organic matter. The carbon dioxide released during bacterial mineralization is readily accessible to microalgae [30,31]. Moreover, bacteria supply the algae partners with fixed nitrogen, various types of vitamin B, and siderophores, while in exchange, the microalgae provide dissolved organic carbon (photosynthates) to bacteria [20,32,33]. Under well-adjusted conditions, the microalgae can produce hydrogen either in axenic mono- or bacterial co-culture systems (e.g., under nitrogen limitation $CO_2$ fixation—the preferred electron sink—is blocked) [7–9]. Anaerobiosis is also essential for the induction of hydrogen production. Anaerobic conditions can be achieved by various approaches, nutrition depletion strategies are the most widely used strategies, however, the addition of actively respiring bacterial partner(s) was also shown to be an efficient and simple solution [5,10,12,16,21,30,34]. A number of studies describe consortial systems where ultimately the bacterial partners produce hydrogen, while the algae components are used to provide specific organic carbon sources to the hydrogen-producing bacteria [5,10,20]. Nevertheless, a few studies have investigated the specific algal hydrogen production in algae–*Bacillus* co-cultures [20,35,36]. *Bacillus* species are widely used in different commercial products utilizing their various direct and indirect plant growth promotion mechanisms. *Bacillus* spp. are often capable of nitrogen fixation, mineralization of phosphorus and other nutrients, phytohormone and siderophore production, generation of antimicrobial compounds and hydrolytic enzymes, and being an inducer of plant systemic resistance and tolerance to abiotic stressors [37–39].

Three different *Bacillus* species were investigated as eukaryotic green algae partners in this experiment. *Bacillus amyloliquefaciens*, *Bacillus mycoides*, and *Bacillus cereus* were all shown to promote the specific algal biomass yield of two *Chlorella* species in co-cultures compared with the yields of the axenic algae under the same growth conditions. *B. amyloliquefaciens* is a non-pathogenic, endospore-forming *Bacillus*, a free-living soil bacterium with a variety of traits, including plant growth promotion, production of antifungal and antibacterial metabolites, and production of industrially important enzymes [40]. *B. mycoides* is a non-motile, spore-forming bacterium able to create rhizoid colonies [41]. *B. cereus* is a common spore-forming and rod-shaped bacterium widely distributed in the environment [42]. Moreover, all three *Bacillus* species are able to produce polysaccharides, which are secreted from the bacterial cells (exopolysaccharides—EPSs). These bacteria are often embedded in their own EPS matrix, creating filament-like biofilm structures [41,43]. These Bacillus-based biofilms are widely defined as multicellular communities occurring at surfaces or interfaces [44].

The specific aim of the present study is to provide a comparative analysis of the biomass and hydrogen production capability of two selected *Chlorella* green algae in various combinations of bacterial co-cultivations. We have investigated the specific effects of three growth-promoting *Bacillus* bacteria on algal hydrogen, biomass, and EPS production.

## 2. Materials and Methods

### 2.1. Growth of Axenic Algal Strains and Pure Bacterial Strains

*Chlorella* sp. MACC-38 and *Chlorella* sp. MACC-360 (both received from the Moson-magyaróvár Algal Culture Collection (MACC)) green algae were used for the experiments. Algae cultures were pre-grown on TAP (TRIS-Acetate-Phosphate) plates at 25 °C under illumination. Algae colonies were harvested from TAP plates and transferred into liquid Minimal Medium (MM). MM (1 L) was prepared in sterile filtered water by adding 1 mL $MgSO_4$ (1 mM), 1 mL $FeCl_3 \cdot CaCl_2$ (1 μM), 10 mL glucose solution (0.4% *w/v*), 5 mL histidine solution (0.0015% *w/v*), 5 mL leucine solution (0.004% *w/v*), 10 mL yeast extract (0.01% *w/v*), and M9 minimal salts (232 g $Na_2HPO_4$, 120 g $KH_2PO_4$, 20 g NaCl, and 40 g $NH_4Cl$, and the pH was set to 7.0). Microalgae were cultured for a period of 4 days in closed 150 mL Erlenmeyer flasks at 25 °C, shaken at 180 rpm, and incubated under 50 μmol m$^{-2}$ s$^{-1}$ light intensity using a light/dark photoperiod of 16 h/8 h. The number of living algal cells was counted with a Luna-FL instrument (Logos Biosystems, Anyang-si, Republic of Korea) using the Fluorescence Cell Counting mode. *Bacillus amyloliquefaciens* (DSM 1060), *Bacillus mycoides* (own isolate), and *Bacillus cereus* (own isolate) were selected to use in the algal–bacterial co-culture experiments. *B. amyloliquefaciens* was grown on LB plates (Luria-Bertani medium) supplemented with 10 g L$^{-1}$ starch at 30 °C, while *B. mycoides* and *B. cereus* were grown on LB plates at 30 °C, then harvested and transferred into liquid MM medium for overnight growth. The number of living bacterial cells was counted with a Quantom Tx$^{TM}$ Microbial Cell Counter (Logos Biosystems).

### 2.2. Algal and Bacterial Co-Cultures in Hypo-Vial Bottles

Algal suspensions were prepared from fresh algal cultures by adjusting the initial concentration of $10^7$ algal cells/mL in fresh MM medium. Each partner bacterium was adjusted to an initial concentration of $10^5$ cells/mL in MM medium. These axenic algal and pure bacterial suspensions were used to establish co-cultures in tightly closed Hypo-Vial serum bottles with a total volume of 40 mL. First, 0.5 mL algal and 50 μL bacterial suspensions were added into the bottles, then 19.45 mL freshly prepared Synthetic Wastewater (SWW) medium was added to the algal–bacterial mixtures to prepare a total of 20 mL co-culture solution in the 40 mL bottles [45]. SWW medium was prepared in 1 L of distilled water by adding the following components: 1.6 g peptone, 1.1 g meat extract, 0.425 g $NaNO_3$ (5 mM), 0.07 g NaCl, 0.04 g $CaCl_2 \cdot 2H_2O$, 0.02 g $MgSO_4 \cdot 7H_2O$, and 0.28 g $K_2HPO_4$; the pH was set at 7.5. All mono- and co-cultures were incubated under 50 μmol m$^{-2}$ s$^{-1}$ light intensity at 25 °C and shaken at 180 rpm using a light/dark photoperiod of 16 h/8 h. All mono- and co-cultures were sampled and analyzed for hydrogen and oxygen levels as well as for the number of living algal cells and bacterial colony-forming units (CFU) every 24 h. All co-culture experiments were performed in three replicates. All calculations, and statistical analyses were performed using GraphPad Prism software version 8.0 for Windows PC (GraphPad Software, San Diego, CA, USA).

### 2.3. Gas Phase Analysis

The hydrogen and oxygen levels in the headspace of the Hypo-Vial bottles were routinely measured using gas chromatography. An Agilent 7890A gas chromatograph (Agilent, Santa Clara, CA, United States) equipped with a thermal conductivity detector and an HP-Molsieve column (length 30 m, diameter 0.320 mm, film 12.0 μm) was used for the hydrogen and oxygen measurements. The temperature of the injector, the TCD detector, and the column were kept at 170 °C, 190 °C, and 60/55 °C, respectively. Samples of 80 μL volumes were analyzed in split mode. Three biological replicates were used for each

measurement. Hydrogen and oxygen calibration curves were used to determine accurate gas volumes. A hydrogen calibration curve was used to determine accurate hydrogen amounts. The following correct formula was used for the conversion of simple GC units: $x = y/239.13$ (x: volume of pure hydrogen gas in μL, y: measured GC unit). The measured yields were normalized for the production of 1 L algae culture: $x = y/239.13 \times 12\,500$. Hydrogen concentrations were measured every day before the gas phase was refreshed (5 min aeration was done by opening the bottles under a sterile hood). Total accumulated hydrogen concentrations were measured every day (without aeration).

## 2.4. Morphological Studies

Confocal Laser Scanning Microscopy (CLSM, Olympus Fluoview FV-300, Olympus Optical Co., Ltd., Tokyo, Japan) was used in this study. 50 μL cultures were collected to Eppendorf tubes and stained with Calcofluor White and/or with Concanavalin A using a final fluorescent dye concentration of 10 μg/μL. After 30 min incubation in dark, the samples (8 μL) were spotted on microscope slides and covered with 2% (*w/v*) agar slice and observed with an Olympus Fluoview FV 1000 confocal laser microscope with 40× magnification objective. Sequential scanning was used to avoid crosstalk of the fluorescent dyes and chlorophyll autofluorescence.

Scanning electron microscopy (SEM) was used to investigate the co-cultures and their EPS production in detail. Cells were fixed with 2.5% (*v/v*) glutaraldehyde and 0.05 M cacodylate buffer (pH 7.2) in PBS overnight at 4 °C. 5 μL of the algal and bacterial suspensions were spotted on a silicon disk coated with 0.01% Poly-L-Lysine. The disks were washed twice with PBS and dehydrated with a graded ethanol series (30%, 50%, 70%, 80%, and 100% ethanol, each for 1 h). Then, the samples were incubated in hexamethyldisilazane, a chemical drying reagent. Chemical-dried samples were coated with 12 nm gold and observed under a JEOL JSM-7100F/LV high-end field emission scanning electron microscope at 250×, 1500×, and 10,000× magnification.

## 2.5. Chlorophyll and Carbohydrates Measurements

Both *Chlorella* algae were grown in SWW medium in Hypo-Vial bottles for two and for four days. For chlorophyll extraction, 1 mL cultures were taken and centrifuged at 13,300 rpm for 10 min. The supernatant was discarded, and 0.5 mL methanol was added to the pellets, which were resuspended with pipetting. The tubes were kept in dark at 45 °C for half an hour, then the samples were centrifuged at 13,300 rpm for 10 min. Absorbance values were measured at 653 nm, 666 nm, and 470 nm using a Hidex microplate reader. Calculations for chlorophyll a, chlorophyll b, and carotenoids were done as described by Lichtenthaler and Wellburn [46].

$Ca = 15.65A_{666} - 7.34A_{653}$

$Cb = 27.05A_{653} - 11.21A_{666}$

$Cx + c = 1000A_{470} - 2.86Ca - 129.2Cb/245$

Where Ca, Cb, and Cx + c are concentrations of chlorophyll a, chlorophyll b, and total carotenoids in $\mu g \ mL^{-1}$, respectively.

For total carbohydrate extraction, pellets obtained after pigment extraction were used. The pellets were washed with Milli-Q water and then further dissolved in 10 mL of Milli-Q water. 1 mL of each sample was taken into a fresh glass tube and 5 mL of anthrone reagent was added. Anthrone reagent was prepared freshly by dissolving 0.5 g of anthrone in 250 mL of concentrated sulfuric acid. After the addition of anthrone reagent, tubes were cooled down and then incubated at 90 °C for 17 min in the water bath. After incubation, tubes were cooled down at room temperature, and absorbance values were taken at 620 nm in a Hidex microplate reader.

## 3. Results

Two *Chlorella* green algae and three selected *Bacillus* species were co-cultured in gastight Hypo-Vial bottles to investigate the specific effects of the bacterial partners on algal

growth, biomass yield, algal biohydrogen production, and biomolecule content, including co-culture-specific extracellular polysaccharides.

### 3.1. Algal Biomass Yield Is Facilitated by Bacterial Partners

All *Chlorella* sp. MACC-38 co-cultures (and also the axenic algal culture) were shown to generate significantly higher (about three times higher) total biomass (wet weight) by the 4th day of cultivation compared with those measured in *Chlorella* sp. MACC-360 algal mono- and algal–bacterial co-cultures (Figure 1A). The maximum total biomass values were reached on the 3rd day in the *Chlorella* sp. MACC-38 co-cultures, while the maximum was reached on the 4th day in the *Chlorella* sp. MACC-360 co-cultures. Counting of the bacterial colony-forming units (CFU) indicated that the partner *Bacillus* species had highly similar growth kinetics in all algal co-cultures irrespective of the specific *Chlorella* partner (Figure 1B). This indicated that the remarkable and differential changes in the total biomass of the algal–bacterial co-cultures were due to the different growth features of the two *Chlorella* species. Even the axenic *Chlorella* sp. MACC-38 had a higher growth rate compared with that of the axenic *Chlorella* sp. MACC-360, and this algal growth rate difference was further increased by the addition of the *Bacillus* partners (Figure 1A,C).

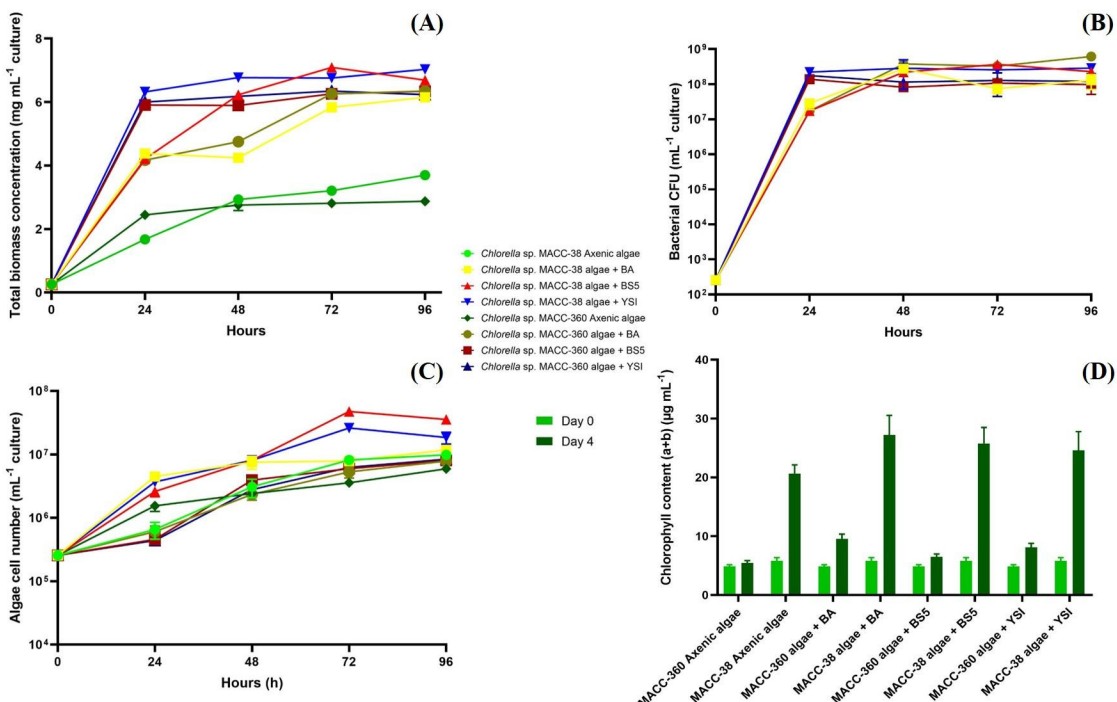

**Figure 1.** Total mono- and co-culture biomass data (**A**), bacterial CFU data (**B**), algae cell number data (**C**), and chlorophyll content (**D**). *Chlorella* sp. MACC-360 and *Chlorella* sp. MACC-38 algal strains were cultivated as axenic mono- and algal–bacterial co-cultures. *B. amyloliquefaciens* (BA), *B. mycoides* (BS5), and *B. cereus* (YSI) were the bacterial partners in co-cultures. Error bars are standard deviations based on three replicates.

*Chlorella* sp. MACC-38 was shown to contain a significantly higher amount of chlorophyll (a + b) than *Chlorella* sp. MACC-360 either in axenic monocultures or in bacterial co-cultures (Figure 1D). This difference was observed both at the start of the experiment (day 0) and also at the end (on day 4). The highest amount of chlorophyll (a + b) was measured in the *Chlorella* sp. MACC-38–*B. amyloliquefaciens* algal–bacterial co-culture on day 4. The images of the culture bottles also confirmed the differences in the chlorophyll (a + b) content between the two *Chlorella* species, as *Chlorella* sp. MACC-38 co-cultures had a much stronger green color compared with *Chlorella* sp. MACC-360 co-cultures (Figure 2).

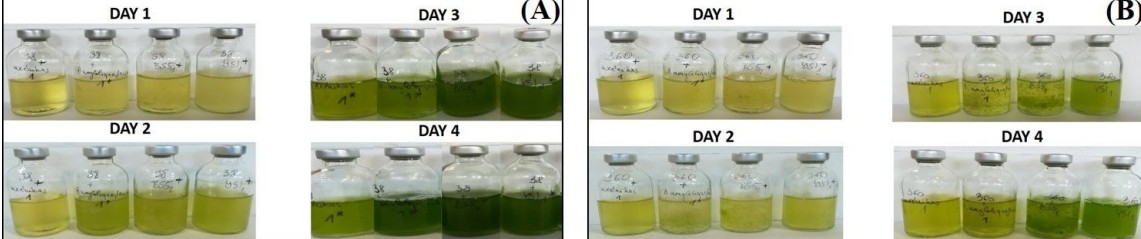

**Figure 2.** *Chlorella* sp. MACC–38 (**A**) and *Chlorella* sp. MACC-360 (**B**) axenic mono- and bacterial co-cultures in Hypo-Vial bottles. Axenic algae and algal–bacterial co-cultures are shown. The first bottles on each day are the axenic algae, the second bottles are the algae co-cultured with *B. amyloliquefaciens*, the third bottles are the algae co-cultured with *B. mycoides*, and the fourth bottles on each day are the algae co-cultured with *B. cereus*.

### 3.2. Co-Culture-Specific EPS Patterns

Scanning electron microscopy (SEM) was applied to investigate the morphology of microalgae and their bacterial partners in monocultures as well as in algal–bacterial co-cultures. SEM analysis revealed that bacterial cells were surrounded by an extracellular matrix in each pure bacterium culture (Figure 3). *B. amyloliquefaciens*, *B. cereus*, and *B. mycoides* bacterial partners all showed extensive cellular aggregation. The cells in the *B. mycoides* cultures especially exhibited thick filament-like structures, which is a known phenomenon [41–44]. All three bacterial partners secreted exopolysaccharides (although to different extents), as revealed by both SEM and confocal laser scanning microscopy (CLSM) (Figures 3 and 6).

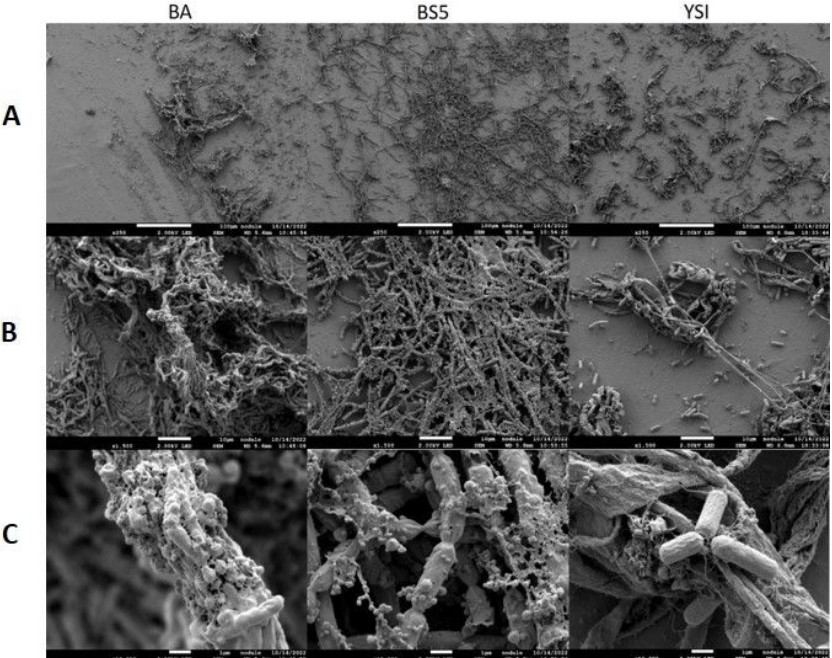

**Figure 3.** SEM analysis of pure bacterial suspensions. *B. amyloliquefaciens* (BA), *B. mycoides* (BS5), and *B. cereus* (YSI) cultures were observed with various magnifications. Images were made on the 2nd day of cultivation. (Panel (**A**)): 250× magnification, scale bars: 100 μm; (Panel (**B**)): 1500× magnification, scale bars: 10 μm; (Panel (**C**)): 10,000× magnification, scale bars: 1 μm.

SEM analysis showed that the axenic algae (both *Chlorella* sp. MACC-360 and *Chlorella* sp. MACC-38) were also embedded in an extracellular matrix produced by the cells (Figures 4 and 5). *Chlorella* sp. MACC-38 was observed to have a slightly larger cell size in axenic culture than *Chlorella* sp. MACC-360 [47]. CLSM analysis revealed that both

axenic algae secreted EPSs that contained α-D glucose (or α-D mannose) sugar residues, as shown by Concanavalin A staining. Specific co-culture morphology patterns were observed when bacterial partners were added to the *Chlorella* species. When *B. amyloliquefaciens* (BA) was added as a bacterial partner, strong algae aggregation was detected in *Chlorella* sp. MACC-360 co-cultures (Figure 4). However, the *Chlorella* sp. MACC-38 cells did not show similar aggregation when co-cultured with *B. amyloliquefaciens* (Figure 5). CLSM analysis revealed another interesting difference between these *Chlorella–B. amyloliquefaciens* co-cultures, more specifically *Chlorella* sp. MACC-360 that algae cells stopped producing the α-D sugar residues in the extracellular matrix, while α-D sugars were detected in *Chlorella* sp. MACC-38–*B. amyloliquefaciens* co-cultures (Figure 6). Another interesting observation regarding the algal cell walls was revealed by CLSM. In co-culture with *B. amyloliquefaciens*, *Chlorella* sp. MACC-360 had a thicker cell wall compared with the axenic MACC-360 alga cells (Figure 6). This was shown by calcofluor white (CFW) staining, which specifically binds to β-D sugar residues. However, the same changes in cell wall thickness were not observed for *Chlorella* sp. MACC-38 co-cultured with the same *B. amyloliquefaciens* partner. When *B. mycoides* (BS5) or *B. cereus* (YSI) were added as bacterial partners to the *Chlorella* algal strains, long, thick multi-cellular filament bundles appeared in the respective co-cultures (Figures 4 and 5).

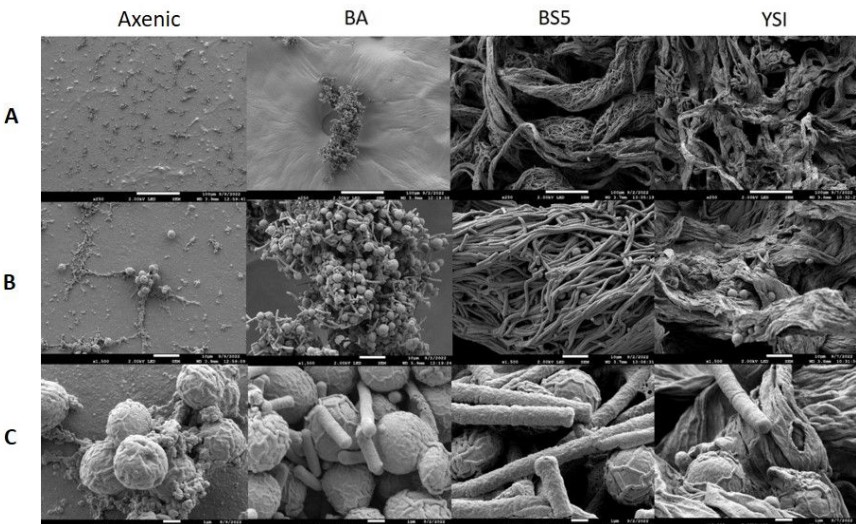

**Figure 4.** SEM analysis of axenic *Chlorella* sp. MACC-360 and its bacterial co-cultures. *B. amyloliquefaciens* (BA), *B. mycoides* (BS5), and *B. cereus* (YSI) were the applied bacterial partners. Images were made on the 2nd day of cultivation. (Panel (**A**)): 250× magnification, scale bars: 100 μm; (Panel (**B**)): 1500× magnification, scale bars: 10 μm; (Panel (**C**)): 10,000× magnification, scale bars: 1 μm.

Interestingly, the addition of either *B. mycoides* (BS5) or *B. cereus* (YSI) as bacterial partners stopped both *Chlorella* algal strains from producing α-D sugar residues in the extracellular matrix, as shown by the lack of green fluorescence in these *Chlorella–Bacillus* co-cultures (Figure 6). However, the appearance of strong blue fluorescence (CFW) in the *Chlorella–B. cereus* co-cultures indicated the presence of β-D sugar residues in the extracellular matrices produced in large quantities by these algal–bacterial communities. Less extracellular CFW fluorescence was detected in the *Chlorella–B. mycoides* co-cultures, and clear differences were found between the *Chlorella* sp. MACC-360–*B. mycoides* and the *Chlorella* sp. MACC-38–*B. mycoides* co-cultures. More specifically, the bacterial filaments were clearly stained by CFW when *Chlorella* sp. MACC-38 was the alga partner, while no bacterial cell wall staining was observed when *Chlorella* sp. MACC-360 was co-cultured with *B. mycoides* (Figure 6).

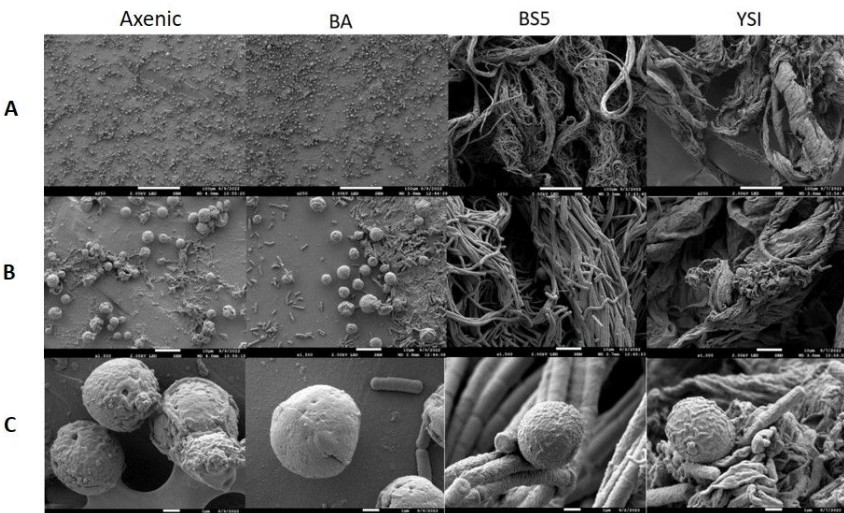

**Figure 5.** SEM analysis of axenic *Chlorella* sp. MACC-38 and its bacterial co-cultures. *B. amyloliquefaciens* (BA), *B. mycoides* (BS5), and *B. cereus* (YSI) were the applied bacterial partners. Images were made on the 2nd day of cultivation. (Panel (**A**)): 250× magnification, scale bars: 100 μm; (Panel (**B**)): 1500× magnification, scale bars: 10 μm; (Panel (**C**)): 10,000× magnification, scale bars: 1 μm.

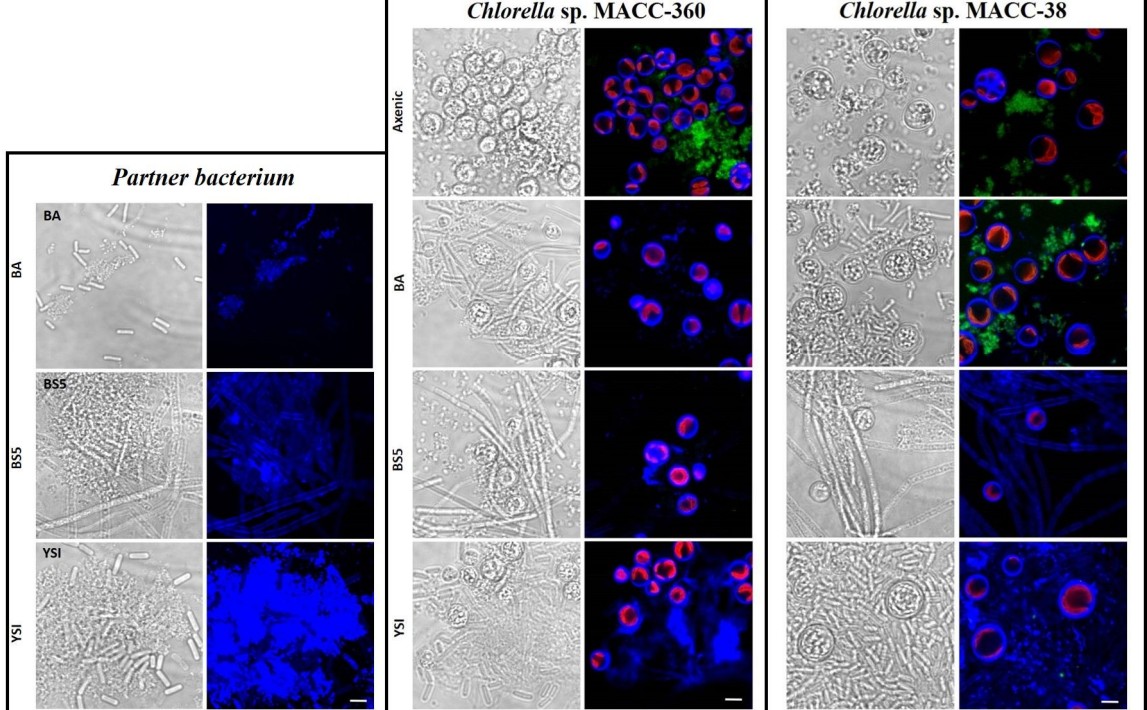

**Figure 6.** CLSM analyses of pure bacteria (*B. amyloliquefaciens* (BA), *B. mycoides* (BS5), and *B. cereus* (YSI)), axenic algal strains (*Chlorella* sp. MACC-360 and *Chlorella* sp. MACC-38), and algal–bacterial co-cultures. All samples were stained with CFW (blue fluorescence specific for β-D sugar residues) and Con A (green fluorescence specific for α-D sugar residues). Chloroplast autofluorescence of live cells is indicated by red color. Scale bars in all pictures represent 10 μm.

### 3.3. Algal Hydrogen Production

The accumulated algal hydrogen production values were influenced by all bacterial partners. *B. amyloliquefaciens* exerted the greatest effect, while *B. mycoides* and *B. cereus* bacterial partners had rather moderate promoting effects on the selected green algae to accumulate hydrogen. Both *Chlorella* algae produced around 40 mL $H_2$ $L^{-1}$ in co-cultures

with *B. amyloliquefaciens*, while only around 25 mL H$_2$ L$^{-1}$ were produced by these algae when the other *Bacillus* partners were applied in the co-cultures (Figure 7A).

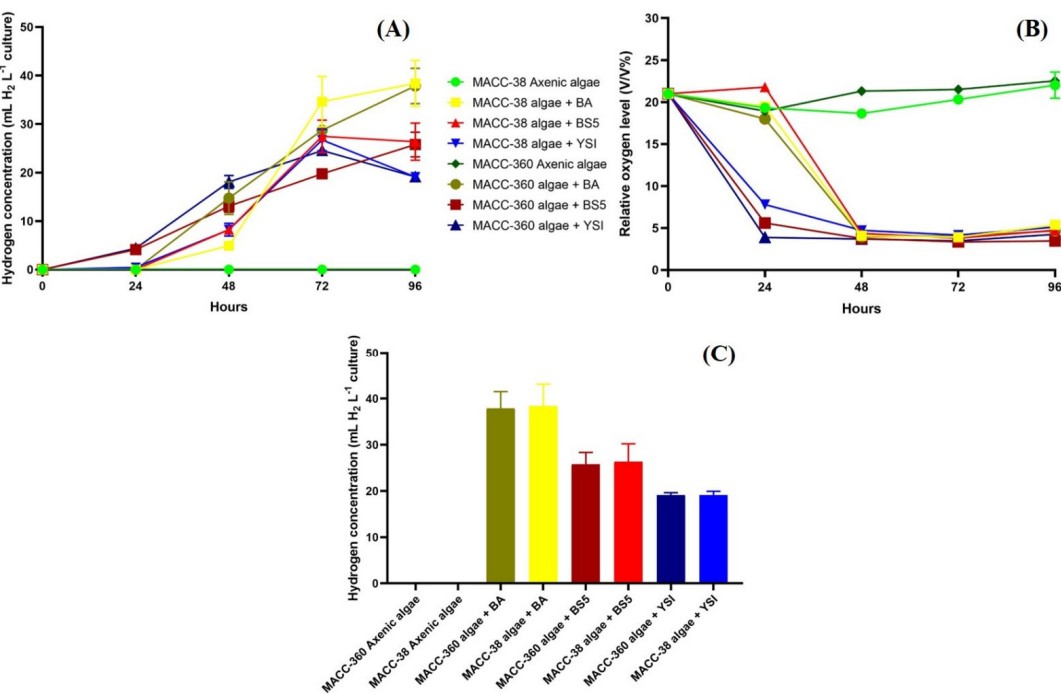

**Figure 7.** Accumulated hydrogen yields measured every day (**A**), dynamics of headspace oxygen concentrations (**B**), and final accumulated hydrogen yields at the end of the experiment (**C**). Headspace hydrogen and oxygen yields of *Chlorella* sp. MACC-360 and *Chlorella* sp. MACC-38 axenic algae and algal–bacterial co-cultures were measured every 24th h. Bacterial partners were *B. amyloliquefaciens* (BA), *B. mycoides* (BS5), and *B. cereus* (YSI). Error bars show standard deviations based on three replicates.

Bacterial respiration was shown to be linked to algal hydrogen production; hydrogen evolved immediately when the oxygen level decreased in the headspace and dissolved when the oxygen level decreased in the solution (Figure 7B). *B. cereus* in both *Chlorella* co-cultures and *B. mycoides* in co-culture with *Chlorella* sp. MACC-360 decreased the headspace oxygen level from 21% to 3% by the end of the 2nd day, while *B. amyloliquefaciens* showed a lower respiration rate when co-cultured with the algal strains. It is important to note that axenic algae cultures maintained high headspace oxygen content due to active photosynthesis and no decrease was observed in the headspace oxygen; therefore, no algal hydrogen production could be detected in the axenic algal cultures (Figure 7A–C).

Daily hydrogen data were measured every 24 h. Again, the axenic *Chlorella* strains did not produce any hydrogen in the SWW medium; this served as a baseline for the co-culture production data (Figure 8.) *Chlorella* sp. MACC-360 produced higher amounts of hydrogen compared with *Chlorella* sp. MACC-38 when co-cultivated with the *Bacillus* bacterial partners (Figure 8). The only exception was observed on the 3rd day: the *Chlorella* sp. MACC-38–*B. amyloliquefaciens* co-culture produced a higher amount of hydrogen compared with its *Chlorella* sp. MACC-360–*B. amyloliquefaciens* counterpart. The maximum daily hydrogen production was observed in the *Chlorella* sp. MACC-360–*B. amyloliquefaciens* co-culture on the 2nd day when a daily hydrogen amount of 26.7 mL L$^{-1}$ culture was measured in the headspace of this algal–bacterial co-culture cultivated in SWW. Since the sum of the daily hydrogen productions exceeded the total accumulated hydrogen amounts measured at the end of the 4th day of cultivation, it can be concluded that accumulated hydrogen inhibited further hydrogen production, which is a known phenomenon [17]. This difference was most visible in the case of the *Chlorella* sp. MACC-360–*B. cereus*

co-culture, where the sum of the daily hydrogen productions (68.9 mL H$_2$ L$^{-1}$ culture) was 3.8 times higher compared with the total accumulated hydrogen (in 96 h) measured in the same co-culture. Although the highest daily hydrogen production was achieved by the *Chlorella* sp. MACC-360–*B. amyloliquefaciens* co-culture, the most balanced daily production was recorded for the *Chlorella* sp. MACC-360–*B. cereus* co-culture, since this combination resulted in a daily average of 22 mL hydrogen with only minor deviations in a 3-day-long period between day 2 and day 4. Similarly, the daily production of the *Chlorella* sp. MACC-360–*B. mycoides* co-culture was quite stable, with a somewhat lower average daily production value of 15 mL over these three days. The generally lower daily hydrogen production level of *Chlorella* sp. MACC-38 was less influenced by its specific *Bacillus* partners.

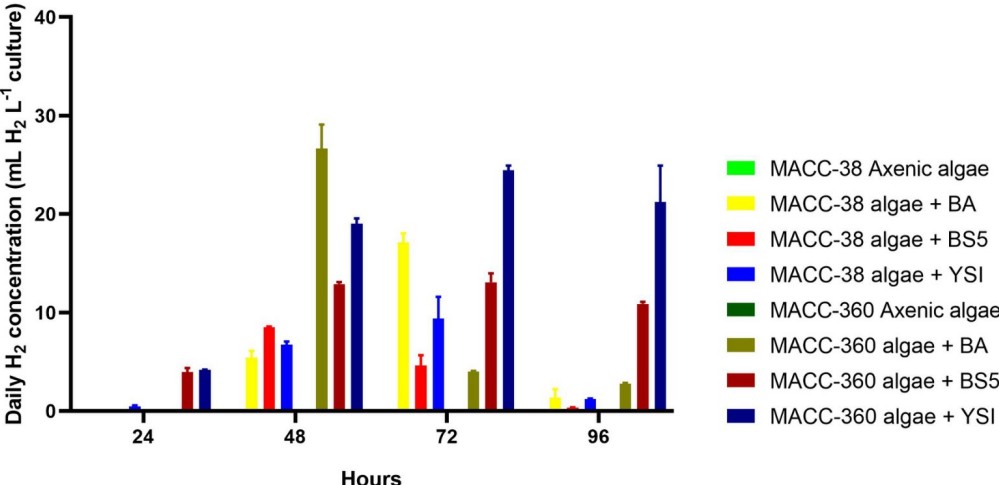

**Figure 8.** Daily hydrogen yields of *Chlorella* sp. MACC-360 and *Chlorella* sp. MACC-38 algae in axenic mono- and bacterial co-cultures. *B. amyloliquefaciens* (BA), *B. mycoides* (BS5), and *B. cereus* (YSI) bacterial partners were applied. Daily hydrogen concentrations in the headspace were measured every 24th h. After each hydrogen measurement, 5 min aeration was applied by opening the bottles under a sterile hood. Error bars show standard deviations based on three replicates.

## 4. Discussion

Interactions between eukaryotic green algae and bacteria are ubiquitous in natural ecosystems, and some of these interactions have been shown to be suitable for utilization in biohydrogen production [5,10,12,34,48]. Using algal–bacterial consortia has a number of advantages over axenic algae or pure bacterial cultures for biohydrogen evolution [10]. Appropriately chosen and applied heterotrophic bacterial partners share nutrients and vitamins with the eukaryotic alga partner and increase algal photosynthetic efficiency by directly respiring the evolved photosynthetic oxygen [33]. The bacterial partners strongly contribute to the maintenance of the anaerobic microenvironment necessary for the activation and function of the algal hydrogenase enzymes. Members of the *Bacillus* genus are among the most studied bacteria; they have essential functions in the soil (e.g., mineralization of phosphorus and other nutrients; phytohormone production; and production of siderophores, antimicrobial compounds, and hydrolytic enzymes) that facilitate plant growth promotion [37–39]. Certain *Bacillus* species were also shown to be suitable for hydrogen production: the bacterial cultures of *Bacillus thuringiensis* strain EGU45 and *Bacillus amyloliquefaciens* strain CD16 produced 2.4–3.0 L H$_2$/day/L during a 60 days continuous culture system [35]. As wastewater is considered a potential substrate for biohydrogen production, both bacterial and algal–bacterial systems have been investigated in synthetic and real wastewater [49–51]. Tests have been conducted in various types of synthetic wastewater (SWW) to investigate the biomass yield and hydrogen productivity potential of various mono- and co-cultures as well as complex microbial communities [20,52,53].

The present study investigated engineered *Chlorella–Bacillus* co-cultures in sterile synthetic wastewater. Two eukaryotic green algal strains were selected for the investigation, *Chlorella* sp. MACC-360 and *Chlorella* sp. MACC-38, both being robust green algae capable of growth in various types of wastewater. *Chlorella* sp. MACC-360 has been investigated in detail previously [54,55], while the characteristics of *Chlorella* sp. MACC-38 have not yet been published. The average cell size values represent the main difference between the two *Chlorella* species, as the average cell size of MACC-38 alga is significantly larger compared with that of MACC-360 when grown in a TAP medium under optimal conditions. As partner bacteria, three different *Bacillus* species were selected for the co-culture experiments in SWW. *Bacillus amyloliquefaciens* (DSM 1060) was used in our previous starch-to-hydrogen conversion study [20], while *Bacillus mycoides* and *Bacillus cereus* were isolated by our research group from soil. Despite the largely different average cell sizes of the two algae strains, no significant differences were observed in the cell number of the algal strains grown in SWW either in axenic or in co-cultures.

Eukaryotic green algae are capable of hydrogen production through their Fe-Fe hydrogenase enzymes. Multiple pathways exist for hydrogen production in microalgae, as described for the model alga *C. reinhardtii* [28,56,57]. The PSII (Photosystem II)-dependent hydrogen production pathway (or direct photolysis) is directly connected to the water-splitting step of photosynthesis in which electrons derived from water splitting are channeled through the whole photosynthetic electron chain to the hydrogenase enzymes. The PSII-independent hydrogen production (also called photofermentation or indirect photolysis) utilizes only PSI (Photosystem I); electrons derived from storage materials (mainly starch) are fed to the plastoquinone pool by the plastoquinone-reducing Type II NAD(P)H dehydrogenase enzyme (NDA2), then go through PSI to the ferredoxin and then to the hydrogenase enzymes. The third possibility for the algae to produce hydrogen is performing simple dark fermentation. In this case, some portion of the fixed carbon (again, mainly starch in green algae) is simply metabolized in a fermentative way and a certain fraction of the electrons derived from pyruvate is directly fed to the ferredoxin through the function of the pyruvate-ferredoxin oxidoreductase [58]. In our present study, the ratios of the various hydrogen production pathways were not investigated.

Differences were detected in the daily hydrogen production patterns of the two algal strains cultivated in SWW. As was expected, the axenic algal cultures did not produce any hydrogen. The presence of bacterial partners was essential to induce hydrogen production of *Chlorella* species [10,20]. *Chlorella* sp. MACC-360 produced a higher daily amount of hydrogen compared with that of *Chlorella* sp. MACC-38 when co-cultivated with any of the bacterial partners. Interestingly, *Chlorella* sp. MACC-38 had a significantly higher chlorophyll (a + b) content compared with that of *Chlorella* sp. MACC-360. This difference was clearly visible both in axenic algae cultures and in algal–bacterial co-cultures throughout the whole period (96 h) of the experiments. This was in agreement with the hydrogen production data, as the higher photosynthetic activity of *Chlorella* sp. MACC-38 resulted in higher algal biomass (Figure 1), while in *Chlorella* sp. MACC-360, a supposedly higher amount of electrons was channeled to the hydrogenase enzymes, and fewer photosynthetic electrons were utilized for carbon fixation through the ferredoxin-NADP-reductase (FNR).

Only minor differences were detected for the two *Chlorella* strains in the accumulated hydrogen production throughout the 96-h-long experiment (Figure 8A). Both algae produced the highest amount of accumulated hydrogen when co-cultured with *B. amyloliquefaciens*, while the least hydrogen was produced by the algae when co-cultured with *B. cereus*. Daily hydrogen production data revealed more specific and interesting differences between the co-cultures (Figure 8). Again, in general, *Chlorella* sp. MACC-360 performed better in daily hydrogen production than *Chlorella* sp. MACC-38. *Chlorella* sp. MACC-360 reached its highest daily hydrogen yield on day 2 and relatively high daily algal hydrogen yields were observed on the following (3rd and 4th) days as well. *Chlorella* sp. MACC-38 showed its highest daily hydrogen production on day 3 (which was still significantly lower compared with the peak of *Chlorella* sp. MACC-360 on day 2), then this alga practi-

cally ceased producing hydrogen, though its biomass increased much faster than that of *Chlorella* sp. MACC-360 (Figure 1). The specific effects of the various *Bacillus* partners on the daily algal hydrogen production were analyzed in detail (Figure 8). The day 2 peak of *Chlorella* sp. MACC-360 was the most evident when the alga was co-cultured with *B. amyloliquefaciens*. However, the most stable and sustainable daily *Chlorella* sp. MACC-360 hydrogen production was achieved when *B. cereus* was the bacterial partner (similarly high algal hydrogen yields were observed every day starting on day 2). The daily hydrogen production of *Chlorella* sp. MACC-360 was also quite stable in co-culture with *B. mycoides*, though this production rate remained stable at a relatively low level. The comparably lower daily hydrogen yields of *Chlorella* sp. MACC-38 were less influenced by the bacterial partners, but the highest MACC-38 hydrogen yields were achieved when *B. amyloliquefaciens* was the applied bacterial partner.

Algal exopolysaccharides can be directly utilized by bacteria for growth, and EPSs are often produced by various bacteria as well [59]. The secreted EPSs might attract or repel other microorganisms and trigger biological responses. The presence of algal EPSs might also help in collecting beneficial bacteria in the environment [47]. Studies investigating the biostimulatory effects of secreted *Chlorella* polysaccharides have shown that these compounds might positively affect bacterial growth and biomass [19].

Interesting and potentially exploitable exopolysaccharide production patterns were also observed in the various co-cultures in our study. Even the axenic *Chlorella* cultures produced EPSs, as was confirmed by fluorescent staining; however, interesting changes were observed in the structure of the generated EPSs in response to bacterial co-cultivations. The EPSs produced by the axenic *Chlorella* strains contained α-d-mannose or α-d-glucose sugar residues since the fluorescent dye Concanavalin A (Con A) possesses a remarkably specific capacity to bind primarily to these residues within the macromolecules [60,61]. The EPSs showing green fluorescence (stained by Con A) and thereby containing α-D sugars was only detected in the *Chlorella* sp. MACC-38–*B. amyloliquefaciens* co-culture. The green fluorescence disappeared in all other algal–bacterial co-cultures. Thus, EPSs with altered structures were identified in the co-cultures of *Chlorella* sp. MACC-38 when cultivated either with *B. cereus* or *B. mycoides* and in the co-culture of *Chlorella* sp. MACC-360 when co-cultured with *B. cereus*. The altered structures of the EPSs were indicated by differential staining in which the extracellular matrices of these specific co-cultures were stained blue with CFW dye, which specifically binds to the β-D glucopyranose polysaccharides within complex macromolecules. It is significant to note that EPSs could not be detected at all in *Chlorella* sp. MACC-360 co-cultures except when *B. cereus* was the bacterial partner. SEM analysis also confirmed the extensive and differential EPS production of the axenic algal strains and certain previously discussed co-cultures.

## 5. Conclusions

Hydrogen is a promising candidate for gradually replacing fossil fuels. Biohydrogen production is still far from being economically feasible; however, intensive research is being conducted on green algae as potential future producers of this clean and environmentally friendly energy carrier. The application of algal–bacterial consortia for algal biohydrogen evolution has a number of advantages over using axenic algal cultures. Two robust *Chlorella* green algal strains in combination with three *Bacillus* bacterial partners were tested for increased hydrogen yield. *Chlorella* sp. MACC-360 in co-culture with *B. amyloliquefaciens* proved to be the most efficient combination in this study. It will be important in the future to investigate the molecular mechanism of algal hydrogen production induced by bacterial co-cultivation. (Meta)transcriptome and metabolome analyses are planned to clarify the contribution of the various algal hydrogen production pathways (photolytic and fermentative pathways). It is important to note that bacterial partners often induce the production of co-culture-specific algal biomolecules. Certain macromolecules (exopolysaccharides in the present study) might be of high relevance in specific further applications, such as plant

biostimulation. Thus, the carefully designed valorization of the specific algal–bacterial biomass can strongly contribute to the economic feasibility of algal biohydrogen.

**Author Contributions:** B.H. composed the manuscript and executed the experiments; G.H. participated in the experimental work; A.F. performed all microscopy analyses; and G.M. designed the study, discussed the literature, and finalized the manuscript. All authors have read and agreed to the published version of the manuscript.

**Funding:** This research was funded by the following international and domestic funds: Lendület-Programme (GM) of the Hungarian Academy of Sciences (LP2020-5/2020) and the Széchenyi Plan Plus National Laboratory Programme (National Laboratory for Water Science and Water Security, RRF-2.3.1-21-2022-00008).

**Institutional Review Board Statement:** Not applicable.

**Informed Consent Statement:** Not applicable.

**Data Availability Statement:** The data presented in this study are available in the main text and in the figures.

**Conflicts of Interest:** The authors declare no conflict of interest.

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
