# Peer review of "Algal Hydrogen Production and Exopolysaccharide Patterns in ChlorellaBacillus Inter-Kingdom Co-Cultures"

_fermentation, doi:10.3390/fermentation9050424_

Round 1

Reviewer 1 Report

Line 44: Rephrase, photosynthesis is always light driven, no light no photosynthesis.

And light does not influence the nutritional value. This is driven by the internal pathways.

Line 56: there are better examples for cocultures and hydrogen production, also for phototrophic heterotrophic cocultures.

Line 64: This is misleading. H2 is just produced if the cells can not fix CO2 (the preferred electron sink) in green algae often induced by nitrogen limitation.

Line 100: MM medium (minimal medium medium) does not make sense, and a recipe would be good.

Line 119 recipe for the SWW medium?

Line 126: are the results of the statistical analysis somewhere shown?

Line 160 for two and for four days. (delete period)

Line 207-209 this is a repetition and can be shortened

Line 213: 30 g/l biomass? This is quite a lot for phototrophic cell suspensions

Figure 1 and 2 can be combined.

Here the results are a little unclear to me, the bacterial cells just grow in the first 24 h but the biomass increased till day four, so the algal cells grow to such a high biomass? This is very unlikely. Also from the chl a value,s here the increase is maximum 2 fold but the biomass increased 5 times? Additionally, you found more chl a as you found biomass, this is impossible. Please explain the results.

Line 282: may be the bacteria cells consume the EPS produced by the algal cells, since it is a good sugar source?

Figure 8: there is no production plotted, it is only a concentration and in (A) it is not clear what kind of time scale is measured.

Also the production is not clear to me. It looks like that the bacteria cells consume all the oxygen and under anaerobic conditions the algae cells produce hydrogen, may be this can be tested, if the axenic cultures are cultivated under anaerobic conditions.

Figure 9, was the gas phase flushed every day? Or is this the accumulated h2 production?

Line 371: why?

 Overall I have a lot of questions regarding this study and it needs more explaination.

First hydrogen in green algae are just possible under anaerobic conditions, here with the bacterial co cultures. Second, i see a clear co2 limitation, since it is not providied in the experiment and not clear to me there it comes from.

thirdly: the discussion should focus a little bit on the basic mechanisms of hydrogen production in green algae, since from work with clamydomonas (as zited from the authors) several mechanisms are clear and they can not be explained with such results. The microscopic pictures are nice and show a clear cellular interaction, this is nice.

Reviewer 2 Report

The manuscript reports the results of biohydrogen production by two Chlorella species from synthetic wastewater in the presence of different Bacillus species. The topic is of great interest at the present time due to the growing demand for environmentally friendly energy sources such as biohydrogen. The manuscript is generally well written and easy to read. The obtained results may bring new knowledge to this area. However, there are some questions about the methodology and calculations. In addition, the discussion of the results can be strengthened, in particular, the resulting facilitation of the production of biohydrogen by algae with the help of bacteria should be better discussed in comparison with similar works. The abstract should contain numerical improvements of H2 production, including H2 yields and production rates.

Specific comments:

Line 22: "bacterial partners were shown to specifically promote algal biomass yield"

How did you differentiate the biomass yields of bacteria and algae in co-cultures?

Lines 104-105: "Bacillus mycoides (own isolate) and Bacillus cereus (own isolate)"

are these isolates deposited at Genbank?

Lines 188-191: "All Chlorella sp. MACC-38 co-cultures (and also the axenic algal culture) were shown to generate significantly higher (about three times higher) biomass by the 4th day of cultivation compared to those measured in Chlorella sp. MACC-360 algal mono- and algal-bacterial co-cultures"

1. How did you measure the biomass yield?

2. How did you differentiate the algae biomass yield from the bacterial biomass yield in co-cultivation?

Lines 298-299: Please specify in the caption, what blue and green colors correspond to (EPS, CW)?

Lines 308-310: "Bacterial respiration was shown to be linked to the algal hydrogen production, hy-308 drogen was evolved immediately when the oxygen level decreased in the headspace and 309 dissolved oxygen level decreased in the solution (Fig. 8 B)." 

1. From Fig. 8B, it can not be concluded that hydrogen was evolved immediately when the oxygen level decreased in the headspace and dissolved oxygen level decreased in the solution.

2. Have you performed comparative experiments with axenic Chlorella and reduced oxygen in headspace, to confirm this thesis?

3. Have you tested the Bacillus species for their hydrogen production activity?

Line 319: "Accumulated hydrogen production (A)"

Wrong expression. Better, Hydrogen yield on day X

Line 319: "headspace oxygen content (B)"

Not the content, but the dynamics of oxygen concentration. In the Y-axis, please change the units

Lines 333-336: "Since the sum of the daily hydrogen productions exceeded the total accumulated hydrogen amounts measured at the end of the 4th day of cultivation, it can be concluded that accumulated hydrogen inhibited further hydrogen production," 

1. Exceeded to what extent?

2. From the fact that the sum of daily hydrogen production exceeded the total accumulated amount of hydrogen measured at the end of the 4th day of cultivation, such a conclusion cannot be drawn. Perhaps some hydrogen has been consumed, or you haven't accounted for headspace pressure.

3. Please show in the materials and methods how you calculated the daily and cumulative production of H2.

Line 425: "Algal exopolysaccharides can be directly utilized by bacteria for growth"

please provide reference

Lines 440-443: What could this indicate?

Reviewer 3 Report

This is a very interesting paper that could be accepted after the following minor revisions:

- Please include more quantitative results in the abstract.

- Please expand the background in the Introduction section. This will help to understand better the novelty of your work.

- Please also double-check for updated related literature. There are some important recent works that have not been discussed.

- Please expand the comparison with respect other works in the results.

- Conclusions must include further future works.

- Check references style and typos.

- Please upgrade English grammar.

Round 2

Reviewer 1 Report

Figure 1:

A: there is a unit missing

D: the chlorophyll content is now mg/ml, which is even higher

Figure 7: hydrogen is still called production, it is no production rate, just the concnetration for a rate a time value is needed

the sampe is true for the figure 8.

spaces are missing for some cases 1ml should be 1 ml

you find several cases in the text

Reviewer 2 Report

Thanks for the improvements. I still have a methodology question:

1. In Section 2.3 you have provided equations for calculation of hydrogen production. 

First of all, GC does not give you results in μL (see y: measured GC unit, unit is μL), but in per cents of H2 in headspace. Secondly, using this equation, you obtained H2 yield in μL/L, however in Figs. 7 and 8, H2 yields are given in mL/L. 

Therefore, please provide the correct equations in the Materials and Methods section to calculate daily and total H2 production considering headspace volume and pressure (pressure will not be atmospheric after H2 production or O2 consumption, you must consider this) as well as GC results. The sum of daily H2 production cannot be that much higher than total H2 production (70ml vs 20ml for Chlorella sp. MACC-360 - B. cereus (YSI) co-culture) if calculated correctly and if H2 is not consumed. Please check.
